# Immunonutrition in Patients with Pancreatic Cancer Undergoing Surgical Intervention: A Systematic Review and Meta-Analysis of Randomized Controlled Trials

**DOI:** 10.3390/nu12092798

**Published:** 2020-09-12

**Authors:** Fu-An Yang, Yang-Ching Chen, Cheng Tiong

**Affiliations:** 1School of Medicine, College of Medicine, Taipei Medical University, Taipei 110, Taiwan; b101103107@tmu.edu.tw; 2Department of Family Medicine, Taipei Medical University Hospital, Taipei 110, Taiwan; melisa26@tmu.edu.tw; 3Department of Family Medicine, School of Medicine, College of Medicine, Taipei Medical University, Taipei 110, Taiwan; 4Division of Gastroenterology and Hepatology, Department of Internal Medicine, Taipei Medical University Hospital, Taipei 110, Taiwan; 5Division of Gastroenterology and Hepatology, Department of Internal Medicine, School of Medicine, College of Medicine, Taipei Medical University, Taipei 110, Taiwan

**Keywords:** immunonutrition, pancreatic cancer, systematic review, meta-analysis

## Abstract

Immunonutrition is administered to improve the outcome of patients with pancreatic cancer undergoing surgery. However, its effect and mechanism of action remain unclear. Therefore, we conducted this systematic review and meta-analysis to assess its effects on postoperative outcome and the immune system. Randomized controlled trials (RCTs) were identified and data extracted by two reviewers independently from electronic databases from their inception to 31 October 2019. The result was expressed as the risk ratio (RR) for categorical variables and mean difference (MD) for continuous variables with 95% confidence intervals (CIs). Six RCTs published from 1999 and 2016, with a total of 368 patients, were included. The results revealed that immunonutrition significantly decreased the rate of infectious complications (RR = 0.47, 95% CI (0.23, 0.94), *p* = 0.03) and the length of hospital stay (MD = −1.90, 95% CI (−3.78, −0.02), *p* = 0.05) by modulating the immune system, especially in preoperative group in subgroup analysis. We therefore recommend that patients with pancreatic cancer undergoing surgery could take the advantage of immunonutrition, especially in the preoperative period.

## 1. Introduction

Pancreatic cancer causes an estimated 45,750 deaths per year, ranking as the fourth leading cause of cancer death in both sexes, and its incidence continues to rise [1]. Recent developments in diagnosis and treatment have improved the outcomes of patients with pancreatic cancer. Surgery still plays a vital role in the treatment of patients with resectable pancreatic cancer, and with advances in surgical techniques, perioperative morbidity and mortality has been decreasing [2]. However, even in some specialized hospitals, the 5-year survival rate remains as low as 15–25% [3,4,5,6]. Moreover, surgery is associated with several morbidities, such as infectious and noninfectious complications, which can prolong the hospital stay [7]. The probable causes of perioperative morbidity include malnutrition and immunosuppression induced by cancer, chemotherapy, and surgical stress. Of these, malnutrition is a critical factor affecting postoperative outcomes [8].

Studies on immunonutrition have indicated that it contains mainly arginine, omega-3 fatty acid, RNA, and other immunomodulating agents. Compared with standard diet, immunonutrition as an enteral supplement may decrease the morbidity and mortality in patients undergoing intra-abdominal surgery [9,10,11,12,13].

Therefore, we conducted a systematic review and meta-analysis of randomized controlled trials (RCTs) to determine the effect of immunonutrition on postoperative complications, length of hospital stay, and the immune system.

## 2. Materials and Methods

### 2.1. Inclusion and Exclusion Criteria

The inclusion criteria of this study were as follows: (1) RCTs; (2) Enrolled patients with resectable pancreatic cancer who underwent the associated operation such as pancreaticoduodenectomy and irreversible electroporation; (3) Intervention and comparison: The trial compared preoperative, perioperative, or postoperative oral supplement of immunonutrition with standard diet. (4) Outcome: Postoperative infectious and noninfectious complications, mortality, length of hospital stay, and immunity.

The exclusion criteria of this study were as follows: (1) included cancer other than pancreatic cancer; (2) not compared with standard diet; and (3) animal experiments.

### 2.2. Search Strategy

The authors independently screened the literature, extracted data, and performed crosschecks according to the Preferred Reporting Items for Systematic Reviews and Meta-Analyses strategy [14]. We searched PubMed and EMBASE databases using the following medical subject heading terms: (“Nutrition Therapy” OR “Nutritional Support”) AND “Pancreatic Neoplasms.” We also searched Cochrane and Google Scholar using the term “immunonutrition” in All Text AND “pancreatic cancer.” The RCTs were identified through the refine search function in the databases, if available. The reference lists of the articles retrieved were manually searched for additional studies. The literature was searched from the date of database inception to 31 October 2019. Two reviewers independently reviewed the full texts of all potentially relevant articles to identify articles meeting the eligibility criteria, and dissimilarities in the decisions were resolved by a third reviewer.

### 2.3. Data Items

For each RCT identified, information regarding the timing of immunonutrition used, brand of immunonutrition, surgical procedure, numbers of participants, comparator groups, and outcome measurements was obtained.

### 2.4. Risk-of-Bias Assessment

The risk-of-bias assessment was performed using the RoB 2 tool, a revised Cochrane risk-of-bias tool for randomized trials [15]. The assessed domains were (1) the randomizing process, (2) deviations from intended interventions, (3) missing outcome data, (4) the measurement of the outcome, (5) the selection of the reported result, and (6) overall bias.

### 2.5. Statistical Analysis

Statistical analysis was performed using RevMan 5.3 software, which was provided by the Cochrane Collaboration. A *p* value of <0.05 was considered statistically significant. We used the *I^2^* test to provide an objective measurement of statistical heterogeneity. According to the Cochrane Handbook for Systematic Reviews of Interventions [16], heterogeneity was quantified using the *I*^2^ statistic with a rough guide for interpretation as follows: 0 to 40%—might not be important, 30 to 60%—may represent moderate heterogeneity, 50 to 90%—may represent substantial heterogeneity, and 75 to 100%—considerable heterogeneity. The random-effects model was used for meta-analysis, and the result was expressed as the risk ratio (RR) for categorical variables and mean difference (MD) for continuous variables, with 95% confidence intervals (CIs) presented.

The funnel plot was not employed to test the publication bias due to the few (<10) studies.

## 3. Results

### 3.1. Search Results

The database search retrieved 58 RCTs. Of them, 21 duplicates were excluded using EndNote X9 [17], and 20 studies, which were not compliant with the inclusion criteria, were excluded after screening their title and abstract. Full-text screening of the remaining 17 studies led to the exclusion of 11:3 with no full text available, 4 multiple publications of the same trial, 2 that included other gastrointestinal cancers, 1 without comparison with a standard diet, and 1 where patients underwent irreversible electroporation. Finally, six articles were selected for this systematic review and meta-analysis [18,19,20,21,22,23] (Figure 1).

### 3.2. Study Characteristics

All included studies were published between 1999 and 2016 and included a total of 368 patients (182 in the intervention group and 186 in the control group). The timing of immunonutrition use was preoperative in two studies [18,19], perioperative in one [20], and postoperative in three [21,22,23]. The immunonutrition used contained mainly arginine, omega-3 fatty acid, RNA, and other immune-modulating agents, and the brands were IMPACT [18,19,20,21,22] and Stresson [23]. The main characteristics of the included RCTs are summarized in Table 1.

### 3.3. Risk-of-Bias Assessment

The quality of the RCTs included were assessed by two reviewers independently by using the RoB 2 tool, a revised Cochrane RoB tool for randomized trials [15]. The risk of bias in each study is illustrated in Figure 2.

Four studies [18,19,22,23] were identified as having low risk in the randomization process, and two was identified as having uncertain risk [20,21]. The risk of deviations from intended interventions was low in five studies [18,19,21,22,23] and uncertain in one study [20]. Six studies were identified as having low risk related to missing outcome data [18,19,20,21,22,23]. Furthermore, two studies were identified as having uncertain risk in terms of outcome measurement [19,21]. Six studies had a low risk for selection of the reported result [18,19,20,21,22,23]. Finally, the risk of overall bias was noted as low in four studies [18,19,22,23] and uncertain in two studies [20,21].

### 3.4. Effect of Immunonutrition on Postoperative Total Complications

Two RCTs [21,22] compared the rates of postoperative total complications. These trials included 212 patients (104 who received immunonutrition and 108 who did not). Pooled analysis revealed no significant difference in the amount of postoperative total complications (RR = 0.79; 95% CI = 0.56, 1.12; *p* = 0.18). The homogeneity across the studies was good (*p* = 0.84, I^2^ = 0%). Subgroup analysis revealed no statistically significant intergroup difference in terms of postoperative infectious complications for preoperative group (not estimable), perioperative group (not estimable), nor postoperative group (RR = 0.79, 95% CI (0.56, 1.12), *p* = 0.18) (Figure 3).

### 3.5. Effect of Immunonutrition on Postoperative Infectious Complications

Three RCTs [14,21,22] evaluated the rates of postoperative infectious complications. These trials included a total of 262 patients (129 received immunonutrition and 133 did not). Pooled analysis revealed a significant difference in the amount of postoperative infectious complications (RR = 0.50; 95% CI = 0.30, 0.84; *p* = 0.009). The homogeneity across the studies was good (*p* = 0.95, I^2^ = 0%). Subgroup analysis revealed a statistically significant intergroup difference in terms of postoperative infectious complications for preoperative group (RR = 0.47, 95% CI (0.23, 0.94), *p* = 0.03), but not for perioperative group (not estimable) nor postoperative group (RR = 0.55, 95% CI (0.26, 1.18), *p* = 0.12) (Figure 4).

### 3.6. Effect of Immunonutrition on Postoperative Infectious Complications—Wound Infection

Four RCTs [18,19,21,22] evaluated the rate of postoperative wound infection. These trials included a total of 297 patients (148 received immunonutrition and 149 did not). Pooled analysis revealed a significant difference in the amount of postoperative wound infection (RR = 0.44; 95% CI = 0.21, 0.91; *p* = 0.03). The homogeneity across the studies was good (*p* = 0.81, I^2^ = 0%). Subgroup analysis revealed a statistically significant intergroup difference in terms of postoperative infectious complications—wound infection for preoperative group (RR = 0.36, 95% CI (0.16, 0.84), *p* = 0.02), but not for perioperative group (not estimable) nor postoperative group (RR = 0.78, 95% CI (0.18, 3.42), *p* = 0.74) (Figure 5).

### 3.7. Effect of Immunonutrition on Postoperative Noninfectious Complications

Three RCTs [19,21,22] compared the rates of postoperative noninfectious complications. These trials included a total of 262 patients (129 received immunonutrition and 133 did not). Pooled analysis revealed no significant difference in the amount of postoperative noninfectious complications (RR = 0.90; 95% CI = 0.66, 1.23; *p* = 0.52). The homogeneity across the studies was good (*p* = 0.92, I^2^ = 0%). Subgroup analysis revealed no statistically significant intergroup difference in terms of postoperative noninfectious complications for preoperative group (RR = 0.88, 95% CI (0.58, 1.34), *p* = 0.56), perioperative group (not estimable), nor postoperative group (RR = 0.93, 95% CI (0.59, 1.23), *p* = 0.75) (Figure 6).

### 3.8. Effect of Immunonutrition on Postoperative Noninfectious Complications—Delayed Gastric Emptying

Three RCTs [19,21,22] evaluated the rate of postoperative delayed gastric emptying. These trials included a total of 262 patients (129 received immunonutrition and 133 did not). Pooled analysis revealed no significant difference in the amount of postoperative delayed gastric emptying (RR = 1.17; 95% CI = 0.59, 2.31; *p* = 0.65). The homogeneity across the studies was good (*p* = 0.71, I^2^ = 0%). Subgroup analysis revealed no statistically significant intergroup difference in terms of postoperative noninfectious complications—delayed gastric emptying for preoperative group (RR = 1.67, 95% CI (0.45, 6.24), *p* = 0.45), perioperative group (not estimable), nor postoperative group (RR = 1.03, 95% CI (0.46, 2.27), *p* = 0.95) (Figure 7).

### 3.9. Effect of Immunonutrition on Postoperative Noninfectious Complications—Fistula Development

Four RCTs [18,19,21,22] evaluated the rate of postoperative fistula development. These trials included a total of 297 patients (148 received immunonutrition and 149 did not). Pooled analysis revealed no significant difference in the amount of postoperative fistula development (RR = 1.00; 95% CI = 0.56, 1.80; *p* = 1.00). The homogeneity across the studies was good (*p* = 0.53, I^2^ = 0%). Subgroup analysis revealed no statistically significant intergroup difference in terms of postoperative noninfectious complications—fistula development for preoperative group (RR = 1.01, 95% CI (0.24, 4.20), *p* = 0.99), perioperative group (not estimable), nor postoperative group (RR = 1.10, 95% CI (0.52, 2.32), *p* = 0.80) (Figure 8).

### 3.10. Effect of Immunonutrition on Postoperative Mortality

Four RCTs [18,19,21,22] evaluated the rate of postoperative mortality. These trials included a total of 297 patients (148 received immunonutrition and 149 did not). Pooled analysis revealed no significant difference in the amount of postoperative mortality (RR = 1.35; 95% CI = 0.27, 6.88; *p* = 0.72). The homogeneity across the studies was good (*p* = 0.51, I^2^ = 0%). Subgroup analysis revealed no statistically significant intergroup difference in terms of postoperative mortality for preoperative group (RR = 0.28, 95% CI (0.01, 6.51), *p* = 0.43), perioperative group (not estimable), nor postoperative group (RR = 2.41, 95% CI (0.36, 16.11), *p* = 0.37) (Figure 9).

### 3.11. Effect of Immunonutrition on Length of Hospital Stay

Three RCTs [18,21,22] recorded the length of hospital stay. These trials included a total of 247 patients (123 received immunonutrition and 124 did not). Pooled analysis revealed a significant difference in the length of hospital stay (MD = −2.03; 95% CI = −3.78, −0.45; *p* = 0.01). The homogeneity across the studies was good (*p* = 0.42, I^2^ = 0%). Subgroup analysis revealed a statistically significant intergroup difference in terms of length of hospital stay for preoperative group (MD = −1.90, 95% CI (−3.78, −0.02), *p* = 0.05) but not for perioperative group (not estimable) nor postoperative group (MD = −3.04, 95% CI (−7.68, 1.61), *p* = 0.20) (Figure 10).

### 3.12. Effect of Immunonutrition on Postoperative Immunity

Two RCTs [20,23] evaluated the effect of immunonutrition on postoperative immunity. A total of 71 patients were included (34 received immunonutrition and 37 did not). The results of the analyses are as follows.

First, pooled analysis revealed no significant difference in CD4+ level on postoperative day (POD) 3 (MD = −1.79; 95% CI = −7.88, 4.30; *p* = 0.56). The homogeneity across the studies was good (*p* = 0.67, I^2^ = 0%). Subgroup analysis revealed no statistically significant intergroup difference in terms of CD4+ level on POD3 for preoperative group (not estimable), perioperative group (MD = −3.10, 95% CI (−11.66, 5.46), *p* = 0.48), nor postoperative group (MD = −0.45, 95% CI (−9.12, 8.22), *p* = 0.92) (Figure 11).

Second, pooled analysis revealed no significant difference in CD4+ level on POD 7 (MD = 2.33; 95% CI = −2.79, 7.45; *p* = 0.37). The homogeneity across the studies was good (*p* = 0.86, I^2^ = 0%). Subgroup analysis revealed no statistically significant intergroup difference in terms of CD4+ level on POD7 for preoperative group (not estimable), perioperative group (MD = 3.00, 95% CI (−6.12, 12.12), *p* = 0.52), nor postoperative group (MD = 2.02, 95% CI (−4.16, 8.20), *p* = 0.52) (Figure 12).

Third, pooled analysis revealed no significant difference in CD8+ level on POD 3 (MD = 3.71; 95% CI = −2.62, 10.04; *p* = 0.25). The homogeneity across the studies was good (*p* = 0.62, I^2^ = 0%). Subgroup analysis revealed no statistically significant intergroup difference in terms of CD8+ level on POD3 for preoperative group (not estimable), perioperative group (MD = 1.70, 95% CI (−8.43, 11.83), *p* = 0.74), nor postoperative group (MD = 5.00, 95% CI (−3.11, 13.11), *p* = 0.23) (Figure 13).

Finally, pooled analysis revealed no significant difference in CD8+ level on POD 7 (MD = 1.30; 95% CI = −5.06, 7.66; *p* = 0.69). The homogeneity across the studies was good (*p* = 0.84, I^2^ = 0%). Subgroup analysis revealed no statistically significant intergroup difference in terms of CD8+ level on POD7 for preoperative group (not estimable), perioperative group (MD = 2.20, 95% CI (−8.47, 12.87), *p* = 0.69), nor postoperative group (MD = 0.80, 95% CI (−7.12, 8.72), *p* = 0.84) (Figure 14).

## 4. Discussion

This systematic review and meta-analysis included seven RCTs (439 patients) that evaluated the effect of immunonutrition on postoperative outcomes, including infectious and noninfectious complications, mortality, length of hospital stay, and immunity. We found significant differences in the following aspects (Table 2):(1)infectious complication, for overall and preoperative group in subgroup analysis(2)wound infection, for overall analysis(3)length of hospital stay, for overall and preoperative group in subgroup analysis

Although we did not find significant differences in CD4+ and CD8+ levels at POD3 or POD7 between the groups, considerable evidence supports the benefit of immunonutrition. Immunonutrition has been demonstrated to modulate immune responses through various mechanisms in patients who have undergone surgery [24,25]. The main components of the immunonutrition used in the included studies were arginine, omega-3 fatty acid, and RNA. Arginine is a beneficial supplement in critically ill patients with effects on vasoregulation and oxygen delivery, but the exact mechanism remains unclear [26,27,28]. Omega-3 fatty acid decreases the rate of infectious complications and shortens the length of hospital stay through its modulation of eicosanoid and cytokine biology [29,30,31]. RNA appears to be beneficial to immune modulation but has been less frequently studied [32]. Taken together, these supplements can have a positive effect on immune system regulation.

Immunonutrition influences the systemic immune system by stimulating the immune system of the bowel (i.e., gut-associated lymphoid tissue) [33,34,35]. Interleukin-1 (IL-1) receptor antagonist (IL-1ra) is a specific inhibitor of the activity of both IL-1α and IL-1β, and can modulate the accompanied inflammatory response [36,37]. Slotwinski et al. [23] found a significant difference when comparing IL-1ra levels in both groups in POD7 and POD10. IL-1ra is the most sensitive marker of postoperative anti-inflammatory response to enteral immunonutrition. The temporary increase in IL-1ra level in immunonutrition group decreases the inflammatory response to extensive surgical trauma and shortens its duration. Immunonutrition can thus help accelerate wound healing and prevent late complications.

Hamza et al. [20] found significant reductions in plasma TNF-α and IL-1α levels, resulting in the amelioration of inflammatory symptoms. In addition, decreased TNF-α and IL-1α levels have a favorable influence on the complement system: a significant reduction in CH50 level and increase in C3 level [38,39]. These changes in the complement system can benefit patients by not only normalizing their inflammatory status but also maintaining normal levels with preservation of the functionality of the complement system. These effects may help prevent severe infections and sepsis, thus decreasing postoperative morbidity and mortality.

Aida et al. [19] concluded that the group receiving immunonutrition had significantly lower PGE2 (prostaglandin E2) levels than the control group. Lower PGE2 levels have an important role in not only decreasing TH1 cell differentiation but also suppressing interleukin-23–mediated TH17 cell expansion [40]. Consequently, this effect may protect the patients against the aggravation of surgical complications, decrease the rate of postoperative infectious complications, and shorten the length of hospital stay.

As stated above, the main components of immunonutrition are arginine, omega-3 fatty acid, and RNA which could modulate the immune system of patients through several pathways. When taken together, these supplements help patients decrease the inflammatory response and accelerate wound healing. It could not only prevent complications from happening but also shorten the length of hospital stay.

According to our analysis, immunonutrition is beneficial when it comes to decreasing infectious complication, wound infection rate, and shortening the length of hospital stay, especially in the preoperative group of our subgroup analysis. It helps the patients modulate their immune systemic preoperatively. Therefore, after they underwent the surgery, the complication rates could be lower by the effect as mentioned above. Furthermore, the length of hospital stay could be decreased. However, there are only data for our analysis in the subgroups of preoperative and postoperative groups. We did not have sufficient data for analysis of the effect of perioperative group. As a result, further studies, including subgroup of perioperative, are warranted.

There are several limitations to our study. First, the included studies administered the immunonutrition to patients at different times. Second, the amount of immunonutrition and duration of its administration were unclear. Therefore, we could not figure the amount and duration needed for its effect according to our analysis. Third, the brand of the immunonutrition supplement was not the same; however, the main components were similar. Finally, we found only a small number of studies focusing on the effects on the immune system; further research is warranted to elucidate these effects.

## 5. Conclusions

This is the first systematic review and meta-analysis that has reviewed the effect of immunonutrition on the immune system in surgical interventions for pancreatic cancer. We found that immunonutrition can significantly decrease the rate of postoperative infections, especially wound infection, and shorten the length of hospital stay. Furthermore, the effect of immunonutrition is significantly obvious in the subgroup analysis of preoperative group. We therefore recommend that patients with pancreatic cancer undergoing surgery take advantage of immunonutrition, especially in the preoperative period. Furthermore, well-designed randomized control trials are required to clarify the effect of immunonutrition.

## Figures and Tables

**Figure 1 nutrients-12-02798-f001:**
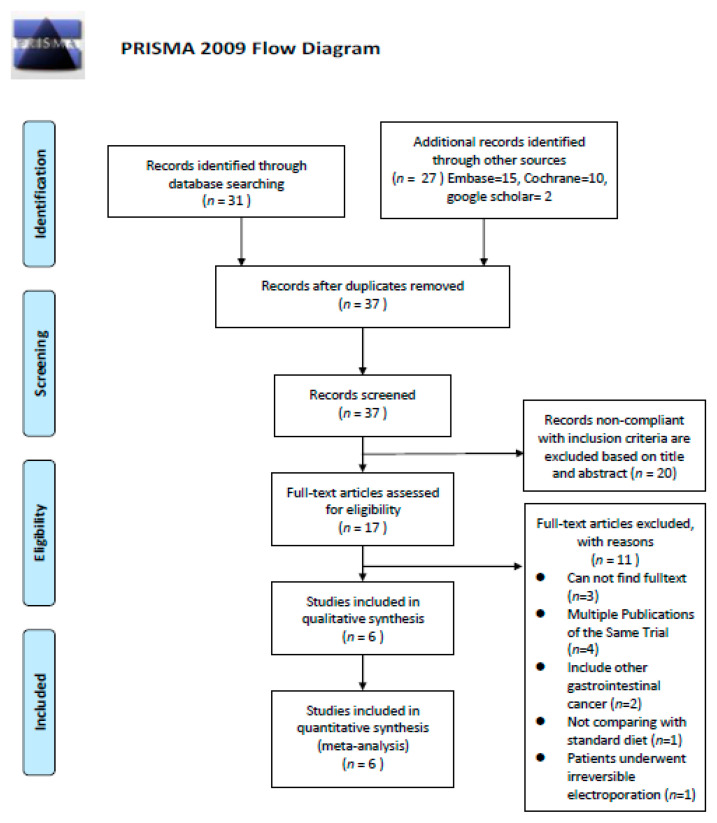
Flow chart showing the details for article inclusion and exclusion.

**Figure 2 nutrients-12-02798-f002:**
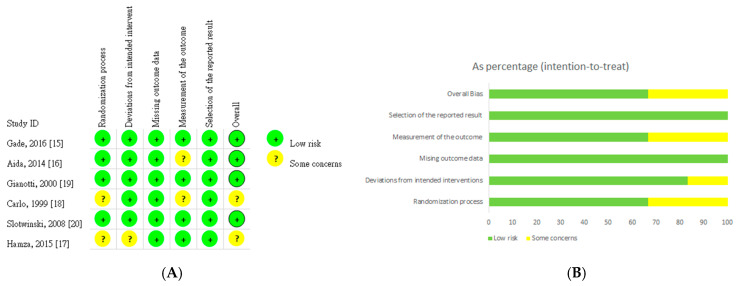
Assessment of risk of bias. (**A**) Risk of bias graph. (**B**) Risk of bias summary.

**Figure 3 nutrients-12-02798-f003:**
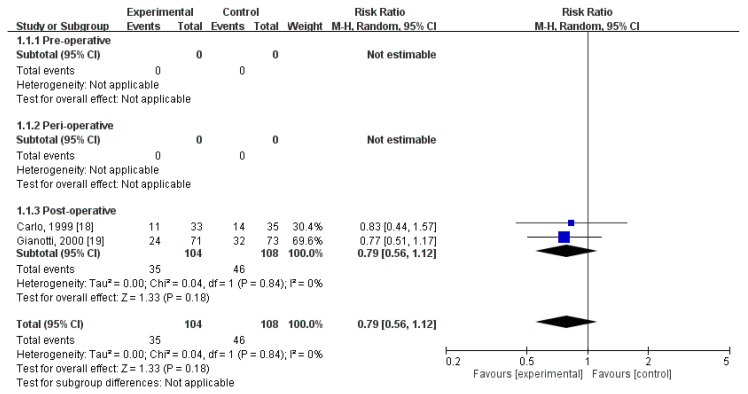
Forest plot of pooled data on postoperative total complications.

**Figure 4 nutrients-12-02798-f004:**
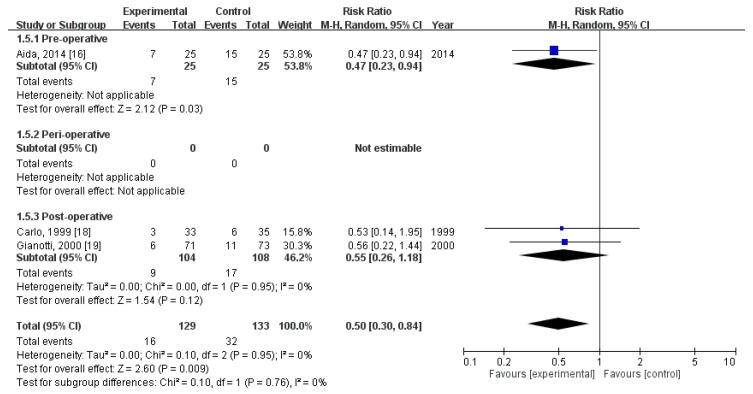
Forest plot of pooled data on postoperative infectious complications.

**Figure 5 nutrients-12-02798-f005:**
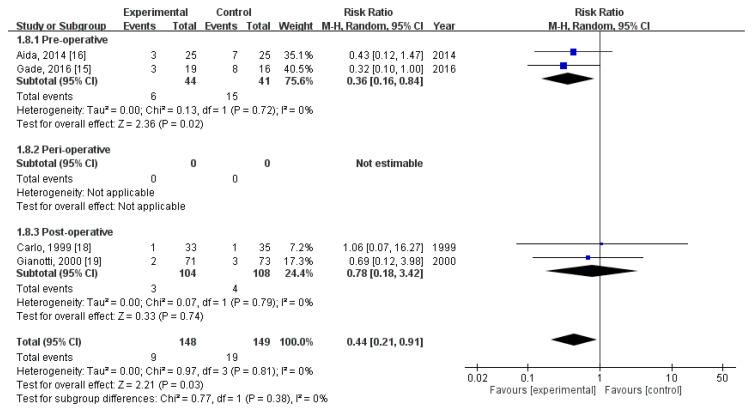
Forest plot of pooled data on postoperative infectious complications—Wound infection.

**Figure 6 nutrients-12-02798-f006:**
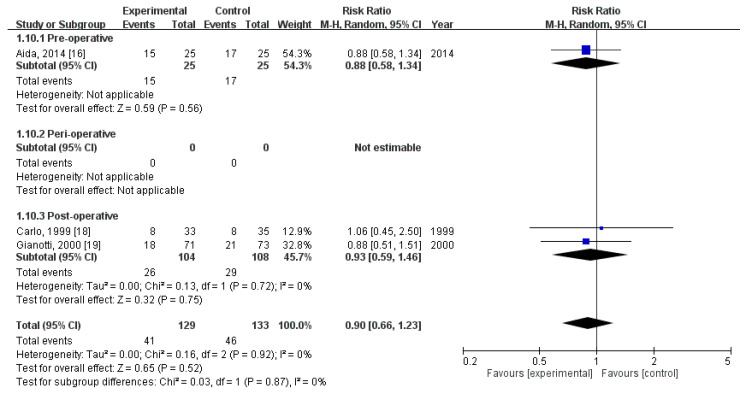
Forest plot of pooled data on postoperative non-infectious complications.

**Figure 7 nutrients-12-02798-f007:**
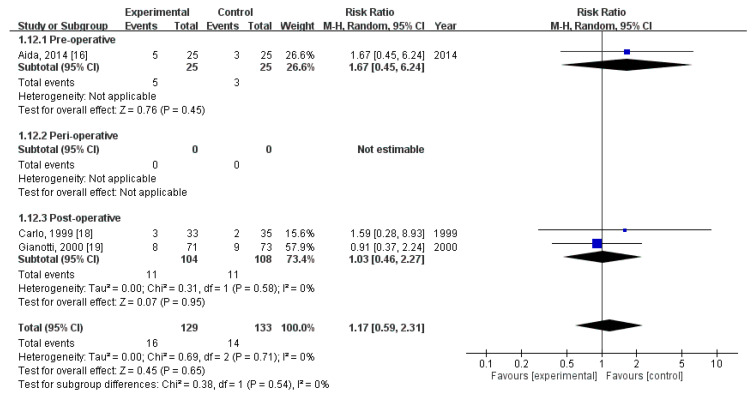
Forest plot of pooled data on postoperative non-infectious complications—Delayed gastric emptying.

**Figure 8 nutrients-12-02798-f008:**
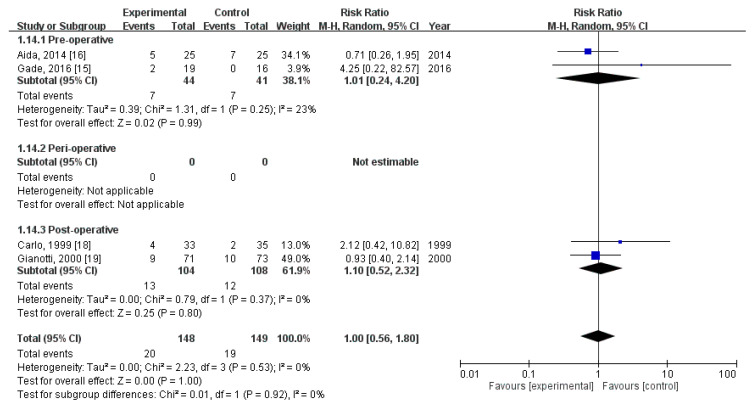
Forest plot of pooled data on postoperative non-infectious complications—Fistula Development.

**Figure 9 nutrients-12-02798-f009:**
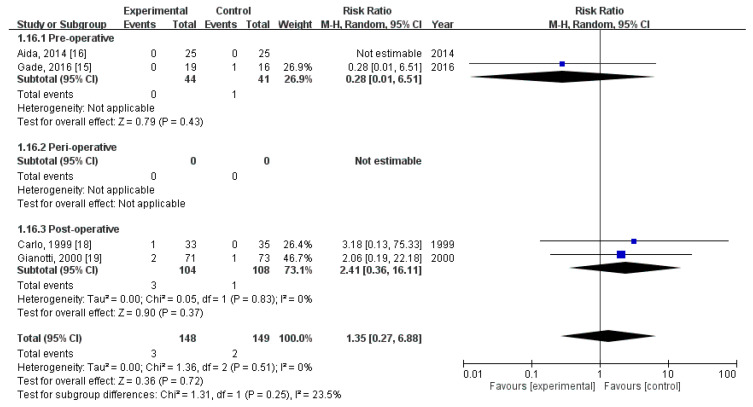
Forest plot of pooled data on postoperative mortality.

**Figure 10 nutrients-12-02798-f010:**
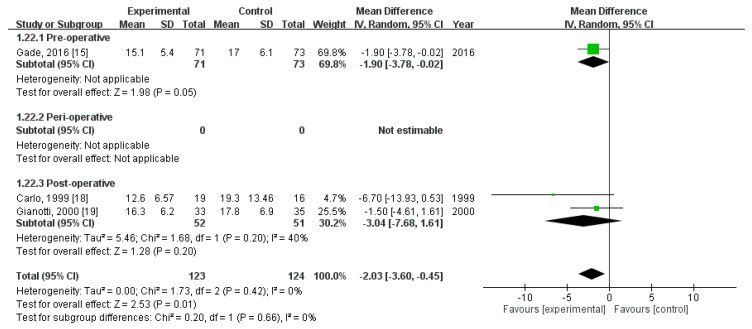
Forest plot of pooled data on length of stay.

**Figure 11 nutrients-12-02798-f011:**
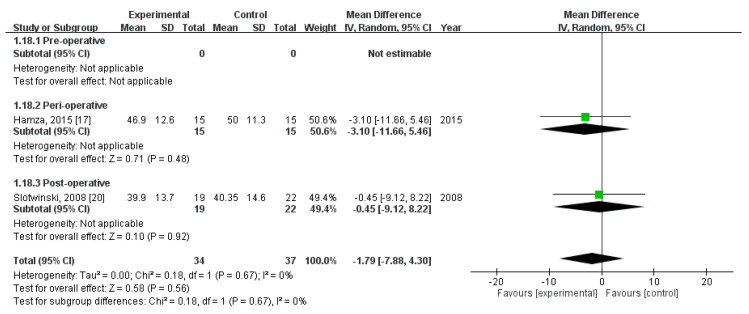
Forest plot of pooled data on CD4+ level on postoperative day 3.

**Figure 12 nutrients-12-02798-f012:**
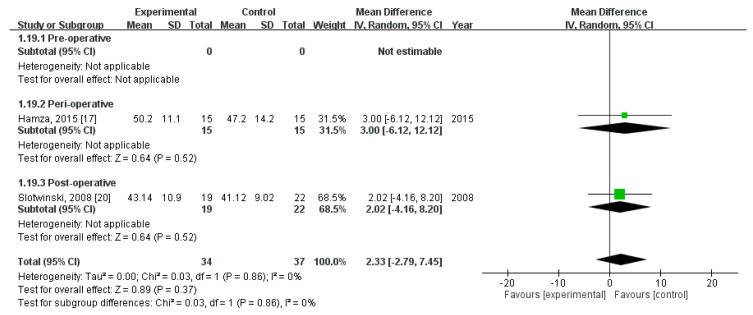
Forest plot of pooled data on CD4+ level on postoperative day 7.

**Figure 13 nutrients-12-02798-f013:**
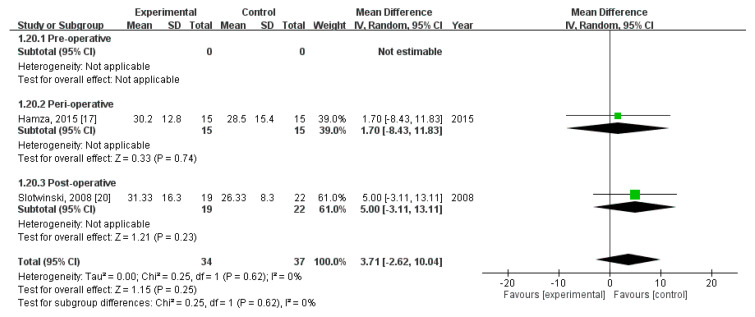
Forest plot of pooled data on CD8+ level on postoperative day 3.

**Figure 14 nutrients-12-02798-f014:**
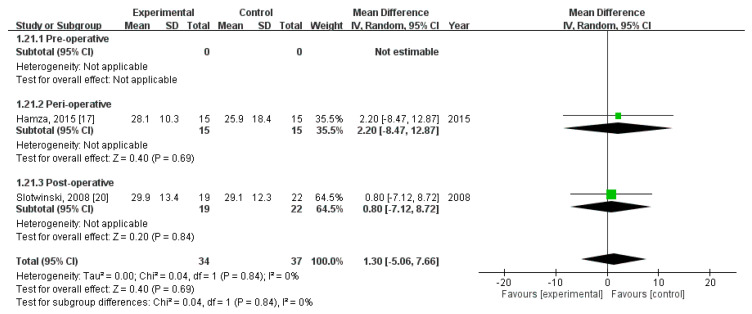
Forest plot of pooled data on CD8+ level on postoperative day 7.

**Table 1 nutrients-12-02798-t001:** Characteristics of included randomized controlled trials.

Author	Year	Time of Administration	*n*	Brand of Immunonutrition	Site of the Tumor	Procedure	Intervention Group *n*	Control Group *n*	Outcome
Gade [18]	2016	preoperative	35	IMPACT	Including pancreatic tail	Curative surgery for pancreatic cancer	19	16	(4)(5)
Aida [19]	2014	preoperative	50	IMPACT	Pancreatic head or peri-ampullary	Pancreaticoduodenectomy	25	25	(2)(3)(5)
Hamza [20]	2015	perioperative	30	IMPACT	Pancreatic head or peri-ampullary	Pancreaticoduodenectomy	15	15	(6)(7)
Slotwinski [23]	2008	postoperative	41	Stresson	Pancreatic head	Pancreaticoduodenectomy	19	22	(6)(7)
Gianotti [22]	2000	postoperative	144	IMPACT	Pancreatic head or peri-ampullary	Pylorus-preserving pancreaticoduodenectomy or Whipple resection	71	73	(1)(2)(3)(4)(5)
Carlo [21]	1999	postoperative	68	IMPACT	Pancreatic head	Pancreaticoduodenectomy	33	35	(1)(2)(3)(4)(5)
Total			368				182	186	

(1) = overall complication, 2 RCTs; (2) = infectious complications, 3 RCTs; (3) = noninfectious complications, 3 RCTs; (4) = length of hospital stay, 3 RCTs; (5) = mortality, 4 RCTs; (6) = CD4+ (%), 2 RCTs; (7) = CD8+ (%), 2 RCTs. RCT, randomized controlled trial.

**Table 2 nutrients-12-02798-t002:** Summary of Subgroup Analysis.

Outcome	Pre-Operative	Peri-Operative	Post-Operative	Overall
Overall complication	-	-	0.79 (0.56, 1.12)	0.79 (0.56, 1.12)
Infectious complication	0.47 (0.23, 0.94) *	-	0.55 (0.26, 1.18)	0.50 (0.30, 0.84) *
Infectious complication—wound infection	0.36 (0.16, 0.84) *	-	0.78 (0.18, 3.42)	0.44 (0.21, 0.91) *
Noninfectious complication	0.88 (0.58, 1.34)	-	0.93 (0.59, 1.23)	0.90 (0.66, 1.23)
Noninfectious complication—delayed gastric emptying	1.67 (0.45, 6.24)	-	1.03 (0.46, 2.27)	1.17 (0.59, 2.31)
Noninfectious complication—fistula	1.01 (0.24, 4.20)	-	1.10 (0.52, 2.32)	1.00 (0.56, 1.80)
Mortality	0.28 (0.01, 6.51)	-	2.41 (0.36, 16.11)	1.35 (0.27, 6.88)
Length of hospital stay (MD, 95%CI)	−1.90 (−3.78, −0.02) *	-	−3.04 (−7.68, 1.61)	−2.03 (−3.60, −0.45) *
CD4+, POD3 (MD, 95%CI)	-	−3.10 (−11.66, 5.46)	−0.45 (−9.12, 8.22)	−1.79 (−7.88, 4.30)
CD4+, POD7 (MD, 95%CI)	-	3.00 (−6.12, 12.12)	2.02 (−4.16, 8.20)	2.33 (−2.79, 7.45)
CD8+, POD3 (MD, 95%CI)	-	1.70 (−8.43, 11.83)	5.00 (−3.11, 13.11)	3.71 (−2.62, 10.04)
CD8+, POD7 (MD, 95%CI)	-	2.20 (−8.47, 12.87)	0.80 (−7.12, 8.72)	1.30 (−5.06, 7.66)

* indicates statistical difference. - indicates not estimable. The results are presented as risk ratio (95% confidence internal) unless otherwise specified. MD, mean difference; CI, confidence interval; POD, postoperative day.

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
