# Peer review of "Immunonutrition in Patients with Pancreatic Cancer Undergoing Surgical Intervention: A Systematic Review and Meta-Analysis of Randomized Controlled Trials"

_nutrients, 2020, doi:10.3390/nu12092798_

Round 1
Reviewer 1 Report
In the revised manuscript, the authors responded sufficiently to the points raised by the reviewers. The manuscript has improved substantially; the results are now well presented and easy to follow. However, there are some points that need attention.
1) It is unusual for tables to be included in the discussion section of the manuscript. However, I could accept this as a recapitulation of the main findings of the study.
2) The discussion is making little use of the main findings and interpretation of the results.
3) The conclusions should be based strictly on the findings of the study and avoid extrapolation. The sentence “… by modulating the immune system through several pathways” should be removed as the study did not look at the actual pathways of immune system modulation. Similarly, based on these findings, preoperative administration of immunonutrition might have some beneficial effects on certain aspects but it cannot be recommended as a routine practice in pancreatic surgery.
4) Fig.1 Flow chart showing the details for article inclusion and exclusion.
Reviewer 2 Report
COMMENTSOn the basis of my comments I douby the paper should be published in the present version
I understand the authors did a huge job with multiple meta-analyses of different outcomes, however the message should arrive clear and simple to the reader.
- From the practical point of view if the authors focus on major pancreatic resections, there is no reason to include the irreversible electroporation and Martin ‘s study should be excluded.
- The main message should be that only preop IN somewhat works in reducing the infection rate and LOS.
- Is there some suggestion for the duration and dosage of IN?
- With regard to this point why there are 2 fig 4 and 5 dealing with the po wound infection?
- It might prove interesting analysing whether some discrepancy in the results is due to different criteria to define surgical complications or fistula and to promote those studies with hard criteria.
- I still consider the discussion (as in my previous review) little pertinent to this study with an excessive emphasis on the immunologic issues which are not significantly emerged in the analysis and could be also deleted to avoid repetitions of concepts which are well known to the readers.
In conclusion, the complexity and abundance of multiple analysis should not at the expense of the clarity of the clinical message.
Round 2
Reviewer 2 Report
The autors have partially answered to the questions
Author Response
Dear editor in chief:
We have done our best in the previous revision. However, there may be some insufficiency of our response. Could you please give us some suggestion so that we could do a better job?
Sincerely,
Fu-An, Yang
This manuscript is a resubmission of an earlier submission. The following is a list of the peer review reports and author responses from that submission.
Round 1
Reviewer 1 Report
The authors present an important review and meta-analysis regarding immuno-nutrtion in pancreatic cancer and pancreatectomy patients. However, there are a number of concerns in this submission.
1) Respectively, the english language is sub-standard. This paper must be reviewed by a English speaking editor, since multiple errors exist in the abstract and introduction alone that are very distracting.
2) The authors should refrain from performing meta-analysis on just 2 or less studies. The risk of Bias is too great and the authors should just publish the data as opposed to trying to over-emphasize a small number of studies.
3) Table 2 is not needed. Redundant for presentation. The authors need to be careful to not "Over Sell' there results given small # of studies, with vary strengths
4) The conclusions are way too over reaching. yes they found positive results but these results maybe found with just "regular" nutrition and that immuno-nutrition may not even be needed.
Reviewer 2 Report
This manuscript needs extensive editing as there are many linguistic and grammar mistakes making it difficult to follow.
The materials and methods section needs a better description, with clear definition of the inclusion and exclusion criteria and the end-points examined.
I wonder if the study by R. Martin (ref 16) should be included in the analysis given that these patients underwent IRE treatment and not a formal pancreaticoduodenectomy
The presentation of the results should be clear, informative and concise.
The discussion is not comprehensive and the possible pathways of immune modulation by immuno-nutrition are not scrutinized.
Despite the detected statistical differences in this meta-analysis, the small number of studies, data heterogeneity and limited information included in the analyzed studies do not provide solid evidence on the beneficial effect of immuno-nutrition.
Reviewer 3 Report
The Yang’s paper “ Immuno-nutrition on patients with pancreatic cancer who underwent surgical intervention - effects and modulation in immune system: A systematic review and meta-analysis of Randomized controlled trials” shows that immune-nutrition reduces the infection rate after pancreatectomy as well as the length of postop stay , hence the authors suggest that it is reasonable to make use of the advantage of immuno-nutrition in patients with pancreatic cancer underwent surgical intervention.
COMMENTS
An important flaws is that the authors did not distinguish among preop IN, postop IN or both preop and postop IN, in such a way which is the practical message to give to the readers? For instance data of Fig 4 would show that only perioperative IN (Aida's study) works in reducing infections, in contrast Fig 10 would show that only preop IN (Gade’s study) works in reducing the length of hospital stay. This also show that there is no consistency about the findings.
Secondly pancreatic surgery includes a major operation, the duodenocephalopancreatic resection associated with very high morbidity and mortality and the distal pancreatectomy, operation much more simple and at lower risk. Absence of stratification by these 2 variables may introduce a major bias in the analysis.
Furthermore the DISCUSSION focuses much more on the potentially beneficial effects of IN and does not go through any interpretation of the data.
For instance postoperative pancreatic fistula is the more relevant and dreaded surgical complication, its rate was not affected by IN , nevertheless the postop stay was shorter in IN-treated arms. Gade, who found a benefit in postop stay, also had a RR 4.25 of fistula favouring the control arm!
All these points warrant a careful reanalysis of the data and an attempt of interpreting the inconsistencies.
Reviewer 4 Report
The manuscript by Yang et al evaluated the role of immuno-nutrition on patients with pancreatic cancer underwent surgical intervention. Overall, this study is interesting and advances our understanding on oral supplement and immune modulation in PC patients receiving surgical intervention. I have some minor concerns:
- The Cochrane library provides standard template for meta-analysis, they should follow their guidelines and reorganize the structure of this manuscript.
- They should include the Funnel plot to show the publication bias.
- They should include the sensitivity analysis.
- They should include the subgroup analysis to show the impact of different intervention time, including preoperative, postoperative and perioperative.
- The title of Fig. 1 is incomplete.
- They should include the subgroup analysis to show the impact of different preoperative nutritional status on the outcomes.
- They should specify whether it’s oral nutrition supplement or intravenous supplement.
- One of the studies included patients that receive irreversible electroporation. Considering irreversible electroporation is totally different from traditional surgery. They should include the sensitivity analysis and subgroup analysis excluding this study.